# Assistive Technology Involving Postural Control and Gait Performance for Adults with Stroke: A Systematic Review and Meta-Analysis

**DOI:** 10.3390/healthcare11152225

**Published:** 2023-08-07

**Authors:** Sujin Hwang, Chiang-Soon Song

**Affiliations:** 1Department of Physical Therapy, Division of Health Science, Baekseok University, Cheonan 31065, Republic of Korea; ptsue@bu.ac.kr; 2The Graduate School of Health Welfare, Baekseok University, Seoul 06695, Republic of Korea; 3Department of Occupational Therapy, College of Natural Science and Public Health and Safety, Chosun University, Gwangju 61452, Republic of Korea

**Keywords:** assistance, cerebrovascular accident, device, postural balance, rehabilitation, walking

## Abstract

This study aimed to comprehensively summarize assistive technology devices for postural control and gait performance in stroke patients. In the study, we searched for randomized controlled trials (RCTs) published until 31 December 2022 in four electrical databases. The most frequently applied assistive technology devices involving postural stability and gait function for stroke patients were robot-assistive technology devices. Out of 1065 initially retrieved citations that met the inclusion criteria, 30 RCTs (12 studies for subacute patients and 18 studies for chronic patients) were included in this review based on eligibility criteria. The meta-analysis included ten RCTs (five studies for subacute patients and five for chronic patients) based on the inclusion criteria of the data analysis. After analyzing, the variables, only two parameters, the Berg balance scale (BBS) and the functional ambulation category (FAC), which had relevant data from at least three studies measuring postural control and gait function, were selected for the meta-analysis. The meta-analysis revealed significant differences in the experimental group compared to the control group for BBS in both subacute and chronic stroke patients and for the FAC in chronic stroke patients. Robot-assistive training was found to be superior to regular therapy in improving postural stability for subacute and chronic stroke patients but not gait function. This review suggests that robot-assistive technology devices should be considered in rehabilitative approaches for postural stability and gait function for subacute and chronic stroke patients.

## 1. Introduction

Gait impairment occurs immediately in more than 80% of patients post-stroke [1]. Notably, most stroke survivors can walk independently without physical assistance from another person; however, 50% of the overall prevalence includes some level of gait impairment [2,3,4]. Gait function after stroke is an essential indicator of functional independence and social integration. Therefore, the proper recovery of gait performance is a vital goal of rehabilitative approach programs for stroke survivors. Consequently, most therapeutic sessions from early stroke rehabilitation are directed at improving gait performance and mobility; however, only 30–50% of stroke survivors are able to participate in community walking [1,2,5].

Despite this, approximately 80% of stroke survivors are capable of independent walking after discharge; most of them show a decreased walking efficacy and abnormal gait cycles, such as a shorter stride length, reduced walking speed, shortened stance phase, and prolonged swing phase of the paretic side, and ultimately an amplified risk of falls and decreased quality of life [2,6]. The most commonly adopted assistive technology devices are walking aids because of gait disturbance, difficulties in daily activities, and social integration for stroke survivors. The assistive technology devices are to increase gait efficiency, reduce residual disability, lower the burden of care, and involve early intensive gait training for stroke survivors in clinical settings [3,7,8].

Mobility assistive devices for patients with a stroke offer the prevention of the fear of falling and risk of a fall, increasing the support surface, improving postural stability and safety, improving functional independence, and improving distance and pace, as well as adaptations around the home [9,10,11]. Mobility assistive devices encompass a range of devices, including walking sticks/aids/crutches that support dynamic balance, wheelchairs that rely solely on mobility, ankle–foot orthosis or slider shoes that improve impairment to specific body parts, and robots that support external skeletons and produce targeted movement [12,13,14,15,16]. Assistive technology devices were initially aimed at assisting patients in increasing repetition and training intensity, as well as maximizing effective intervention time. However, they have now evolved to also support therapy professionals in their work. Given the current situation of limited skilled therapists and a growing number of patients requiring mobility rehabilitation, the purpose and nature of assistive technology are shifting to alignment with technological advancements [17,18,19]. For instance, once the therapeutic effectiveness of body weight support training has been proven, robot-assistive gait training will be developed as a means to eliminate the need for manual assistance from therapists and to facilitate continuous body weight support training [19,20,21].

Of the most importance when choosing an assistive device is its effectiveness for patients with a stroke. A suitable assistive device must be selected carefully, considering the patient’s endurance, cognitive function, strength, and environmental needs. The suitable assistive devices according to the purpose of rehabilitation and the verification of their effect should be progressed in line with technological development [12,19]. On the other hand, there is a lack of research on the preferred aids and their impact on the recovery stage after a stroke. This review aimed to present clinical evidence regarding the most frequently utilized assistive devices and to provide a comprehensive summary of the therapeutic effects of assistive technology devices on postural control and gait function for stroke patients. Furthermore, a meta-analysis was conducted to establish evidence on the effects of commonly used assistive technology devices in improving postural control and gait function for stroke patients.

## 2. Materials and Methods

### 2.1. Review Design

This review’s protocol that followed the Preferred Reporting Items for Systematic Review and Meta-analysis (PRISMA) guidelines was registered in PROSPERO (Registration Number: CRD42023428911) [22]. According to the protocol, two assessors (S and C) independently consulted with each other to clarify the eligibility of the studies, and their collective judgment determined whether the records were deemed appropriate for inclusion. A final decision on study inclusion was made, followed by data collection. The results of the two assessors were compared following the Peer Review of Electronic Search Strategies 2020 Guidelines and an evidence-based checklist. We then synthesized these studies quantitatively and qualitatively.

### 2.2. Eligibility Criteria

This study involved randomized controlled trials (RCTs) that investigated the therapeutic effects of assistive technology devices on postural control and gait performance. The inclusion criteria were as follows: (1) focus on assistive-technology-based training, (2) participants diagnosed with a stroke between 18 and 85 years of age without other neurological diseases, (3) the inclusion of patients in one of the three recovery periods (acute, ≤3 weeks; subacute, between 3 weeks and ≤6 months; and chronic stroke, >6 months), (4) the utilization of core outcome sets to measure postural control and gait performance, (5) involving human participants, (6) written in English, and (7) published as full reports. Included records were excluded if they were (1) unrelated to assistive technology device-based approaches, (2) focused on other neurological disorders except stroke, (3) not focused on postural stability and gait function training, (4) non-human studies, (5) non-original studies, such as news, notes, reviews, opinion pieces, letters, editorials, and comments, and (6) considered gray literature, such as unpublished dissertations, conference supplies, and research abstracts.

### 2.3. Information Sources and Search Strategy

All records published up to 31 December 2022 were searched in four electronic databases: PubMed, Medline, Embase, and ProQuest. The search strategy was a combination of the following medical subject heading and related terms: (Stroke OR Cerebrovascular accident OR Brain vascular accident) AND (Assistive technology OR Assistive device*) AND (Balance OR Postural control OR Postural stability OR Walking OR Gait OR Locomotion) AND (Randomized controlled trial* OR Randomized controlled trial*). This study also reviewed the reference lists of all the identified relevant publications.

### 2.4. Screening of Searched Studies

After searching and compiling the retrieved studies, duplicate studies were removed using a bibliographic information management program based on the studies’ digital object identifier, journal title, volume and issue, and pages. The list of searched studies was cross-referenced to identify and eliminate duplicates. The selection process of this study consisted of two stages: stage I involved screening based on titles and abstracts to eliminate irrelevant reports; stage II was based on full reports to determine inclusion.

Given the focus on stroke, assistive technology devices, postural stability, and gait function, two assessors independently searched and assessed the suitability of the selected studies. The studies were screened based on their titles and abstracts to determine whether they met the inclusion or exclusion criteria. In cases where it was challenging to select based solely on the abstract, the researcher retrieved and reviewed the full text of the article. If the selection process was ambiguous, the final decision was not based solely on the judgment of a single researcher.

To be included in the meta-analysis, the selected studies had individual participant data, including the number of participants, the values of the mean, and standard deviation, because the standard approach to a meta-analysis of continuous outcomes requires information on the mean and either the standard deviation (SD), variance, or standard error values for each group [23]. If the studies lacked individual participant data, we found an alternative method of deriving the missing values of the mean or SD from median and quartile values [24,25]. However, we excluded the records from the meta-analysis when it was impossible.

### 2.5. Data Extraction

To summarize the evidence of assistive technology devices on postural stability and gait performance for stroke patients, this review extracted the following information from each selected study: the authors, year of publication, nation, population and age of participants, intervention details, device types, therapeutic intensity, comparison group, outcome, additional therapy provided, and summary of findings, if applicable.

To ensure quantitative consistency, we calculated the SD using the 95% confidence interval (CI) in selected studies that presented the mean and 95% CI. In cases where a study did not provide the means and SDs of the outcome measures, we calculated the values using the median and interquartile range from the selected studies that presented the median and quartile values [24,25,26]. A review protocol was established for data extraction to minimize selection bias. The protocol included selecting a sample and listing all items to be extracted from the primary RCTs. Two assessors independently performed the data extraction process. Any discrepancies or uncertainties were resolved through consultation with an expert to ensure consistency and accuracy. A final decision regarding the extracted data was made based on this collaborative process.

### 2.6. Data Analysis

The reviews were analyzed using a RevMan 5.4.1 program, which was provided at http://ims.cochrane.org/revman accessed on 19 May 2021. The mean and SD values were combined to calculate the mean difference and 95% CI to assess the effect estimates of the selected RCTs. Heterogeneity was assessed using the I^2^ statistic to ensure accurate interpretation and provide meaningful conclusions for clinical decision making. When at least three studies had relevant data and a sufficient homogeneity in the population, therapeutic interventions, and outcome measures, a meta-analysis was executed using a random-effects model [27]. An I^2^ > 40% threshold was used to determine statistical heterogeneity, and random effects’ models were employed in such cases [23]. Subgroup analyses examined the effects on postural stability and gait performance. If a study had one control group and two experimental groups with identical findings, the data from the experimental groups were pooled during the synthesis used by the RevMan program. Ten RCTs (five for the subacute phase and five for the chronic subacute phase) were investigated throughout the meta-analysis.

## 3. Results

### 3.1. Literature Characteristics of Included RCTs

This study identified 1065 records from four databases, including PubMed (*n*=138), Embase (*n* = 123), Medline (*n* = 80), and ProQuest (*n* = 724). All of them included 291 duplicated records. After excluding the duplicated records, 774 records were screened based on the titles and abstracts and then the following 730 records were excluded: non-RCTs (*n* = 82), non-stroke studies (*n* = 194), and different PICO (*n* = 454). A total of 44 reports were sought for full-text retrieval, except for one record (not reporting participant’s information), leaving 43 reports assessed for eligibility. In total, 13 studies were excluded for the following reasons: being non-RCTs (*n* = 7), being different PICO (*n* = 3), and having the same data reported (*n* = 3). Finally, 30 studies were included in the qualitative synthesis, consisting of 12 and 18 RCTs in the subacute and chronic stages, respectively, based on patients’ recovery periods (see Figure 1).

### 3.2. Qualitative Synthesis of Selected RCTs to Review Postural Control and Gait Performance in Stroke Survivors

This review selected twelve RCTs involving patients with a subacute stroke [3,16,28,29,30,31,32,33,34,35,36,37]. The study involved 460 subacute stroke patients who received robot-assistive training, trunk stabilization exercise, or conventional walking training. They used ankle–foot orthosis (AFO), a compressor belt, or a robot gait training system (assist-as-needed principles and multiple degrees of freedom, an E-go device, an anklebot, a morning walk, a regent suit, an i-walker, and a Lokomat). The studies of the robot-assistive training provided 30 min to 2 h per session (10 sessions to 30 sessions), and the studies of the AFO provided 14 sessions in 26 weeks. The RCTs of the trunk stabilization exercise with a pelvic compressor belt provided 30 min per session (30 sessions). They used the joint angle, a sensory organization test (SOT), a 10 m walking test (10MWT), a 6 min walking test (6mWT), the Barthel index (BI), the Berg balance scale (BBS), the center of loading (COL), the functional ambulation category (FAC), FGA, the Fugl-Meyer assessment of motor functioning—lower extremity (FMA-LE), the modified Ashworth scale (MAS), the motricity index (MI), the limit of stability (LOS), the Rivermead mobility index (RMI), a timed up-and-go test (TUG), a stairs test, Tinetti’s scale, spatiotemporal parameters in a gait analysis (gait cycle duration, single-support period, step length and width, cadence, and gait speed), and others in a gait analysis (moment, power, pelvic tilt, thoracic tilt, ground reaction times, and the center of pressure progression). Four RCTs reported more benefits of robot-assistive training than comparison therapy [31,33,34,35], and only one RCT reported more positive effects of AFO provision than comparison therapy (Table 1 and Table 2) [30].

The review selected 18 RCTs involving patients with a chronic stroke (Table 3 and Table 4) [38,39,40,41,42,43,44,45,46,47,48,49,50,51,52,53,54,55]. They involved 639 chronic stroke patients. These RCT records included 12 types of robot-assisted training using Lokomat, Ekso, RoboGait, an electrical walker, a wearable hip-assistance robot, a trunk stability rehabilitation robot trainer, Exowalk, a GEAR system, and a Bionic leg device to improve the gait function for stroke patients [38,39,42,44,46,47,48,49,50,51,52,54]. The robot-assisted training involved 30 min to 60 min per session and 12 to 40 sessions. The RCTs with robot-assisted training used the activities-specific balance confidence scale (ABC), BBS, 10MWT, 6mWT, RMI, TUG, MAS, FAC, K-MBI, MI, FMA-LE, five-time sit-to-stand test (5XSST), California functional evaluation 40 (CAFÉ 40), Emory functional ambulation profile (EFAP), fall efficacy scale (FES), functional reach test (FRT), global rating of change (GRC), GQI, gait quality index (GQI), Rivermead visual gait assessment (RVGA), 36-item short-form survey (SF-36), medical outcome study 8-item short-form health survey (SF-8), Romberg test, stroke impact scale (SIS), visual analog scale (VAS), electromyography, electroencephalography, and spatiotemporal parameters in a gait analysis. Seven RCTs reported more benefits of robot-assistive training than comparison therapy [38,39,42,46,50,51].

They also included six other types of training, including a toe spreader, an ankle movement system, postural insoles, a foot drop stimulator, non-elastic taping, and an ankle stretcher to improve the gait performance for stroke patients [40,41,43,45,53,55]. They also measured MAS, BBS, proprioception, the range of motion, strength, FMA-LE, TUG, FES, 6mWT, FRT, SIS, FAC, RMI, 10MWT, MI, MBI, SOT, and spatiotemporal parameters in a gait analysis. However, they did not include any assistive technology devices to improve postural control (only for patients with a chronic stroke). Four RCTs reported more positive effects of therapeutic intervention than comparison therapy (Table 2) [41,43,45,55].

### 3.3. Effectiveness of Assistive Technology Device on Postural Control and Gait Performance in the RCT Studies

In patients with a subacute stroke, three RCTs involving 110 patients, providing the participants’ information, assessed the gait velocity [3,28,34]; four RCTs involving 183 patients, providing the participants’ information, assessed the BBS [3,32,33]. Three RCTs involving 90 patients, providing the participants’ information, assessed the FMA-LE score [3,28,32], and three RCTs involving 123 patients, providing the participants’ information, assessed the FAC score [3,32,33]. The scores for BBS are significantly different between the experimental and control groups. The total mean difference (95% CI) values were 3.98 (1.19, 6.77), and the heterogeneity values were Tau^2^ = 0.00, Chi^2^ = 1.79, df = 3 (*p* = 0.62), and I^2^ = 0% for BBS. The test for the overall effect yielded Z = 2.80 (*p* = 0.005) for BBS. However, the other three measures did not differ significantly between the experimental and control groups. The total mean difference (95% CI) values were as follows: 0.10 (−0.05, 0.26) for gait velocity, −1.18 (−4.67, 2.51) for FMA-LE, and −0.14 (−0.64, 0.36) for the FAC score. The heterogeneity values were as follows: Tau^2^ = 0.01, Chi^2^ = 4.64, df = 2 (*p* = 0.10), and I^2^ = 57% for gait velocity; Tau^2^ = 3.98, Chi^2^ = 3.22, df = 2 (*p* = 0.20), and I^2^ = 38% for FMA-LE; and Tau^2^ = 0.02, Chi^2^ = 2.23, df = 2 (*p* = 0.33), and I^2^ = 10% for the FAC score. The test for the overall effect yielded Z = 1.33 (*p* = 0.18) for gait velocity, Z = 0.59 (*p* = 0.56) for FMA-LE, and Z = 0.54 (*p* = 0.59) for the FAC score. A random-effects model was selected because of significant heterogeneity (Figure 2).

In patients with a chronic stroke, five RCTs involving 152 patients provided the participants’ information and assessed the BBS score [38,47,48,49,52], three RCTs involving 81 patients provided the participants’ information and assessed the TUG score [47,50,52], five RCTs involving 135 patients provided the participants’ information and assessed the 6mWT score [44,48,49,50,52], and three RCTs involving 110 patients provided the participants’ information and assessed the FAC score [47,48,49]. The scores for BBS are significantly different between the experimental and control groups. The total mean difference (95% CI) values were 2.55 (0.96, 4.14), and the heterogeneity values were Tau^2^ = 0.00, Chi^2^ = 3.00, df = 5 (*p* = 0.56), and I^2^ = 0% for BBS. The test for the overall effect yielded Z = 3.15 (*p* = 0.002) for BBS. The scores for FAC were also significantly different between the two groups. The total mean difference (95% CI) values were 0.40 (0.04, 0.77), and the heterogeneity values were Tau^2^ = 0.00, Chi^2^ = 1.96, df = 2 (*p* = 0.37), and I^2^ = 0% for the FAC score. The test for the overall effect yielded Z = 2.17 (*p* = 0.03) for the FAC score. However, the other two measures did not differ significantly between the experimental and control groups. The total mean difference (95% CI) values were −2.79 (−9.36, 3.77) for TUG and 1.26 (−26.33, 28.85) for 6mWT. The heterogeneity values were Tau^2^ = 2.15, Chi^2^ = 2.13, df = 2 (*p* = 0.35), and I^2^ = 6% for gait velocity, as well as Tau^2^ = 0.00, Chi^2^ = 1.14, df = 4 (*p* = 0.89), and I^2^ = 0% for 6mWT. The test for the overall effect yielded Z = 0.83 (*p* = 0.40) for TUG and Z = 0.09 (*p* = 0.93) for 6mWT. A random-effects model was selected because of significant heterogeneity (Figure 3).

## 4. Discussion

This review aimed to present clinical evidence regarding the most frequently utilized assistive devices and to provide a comprehensive summary of the therapeutic effects of assistive technology devices on postural control and gait function for stroke patients. The main results of this review are as follows: First, the most frequently applied assistive technology devices for postural stability and gait function were robotic technology for subacute and chronic stroke patients. Other assistive technology devices were pelvic compressor belts and AFO for subacute stroke patients and a toe spreader, an ankle movement system, postural insoles, non-elastic taping, and an ankle stretcher for chronic stroke patients. Second, assistive technology training with robotics significantly benefits postural stability compared with conventional therapy in subacute patients [3,28,32] and chronic patients [47,48,49,50,52]. However, assistive technology training with robotics demonstrates a significant positive benefit on gait function compared with conventional therapy in subacute patients [3,28,32,34,56] and in chronic patients [44,47,50,52], even though chronic stroke patients showed a decreased assistive degree measured with FAC [47,48,49]. Third, the BBS score significantly differed between the experimental and control groups in subacute and chronic patients. However, gait velocity, FAC, and FMA-LE for subacute stroke patients and TUG and 6mWT for chronic stroke patients were not significantly different in the experimental group compared to those of the control group.

Robot-assistive devices are classified into two different types, exoskeletons and end-effectors [57,58]. The exoskeleton type is an external skeleton attached to the lower limbs and actuated by motors to create stepping movements [59]. The RCTs in this review conducted the exoskeletons to support standing posture and dynamic balance during stepping movement for subacute stroke patients [3,16,28,32,34,35] and for chronic stroke patients [38,39,42,44,46,47,48,49,50,51,52,54]. The exoskeleton robots are classified into three types: robotics with a body-weight-supported system [16,28,32]; a whole-body suit type covered with a vest, shorts, knee caps, and foot straps [34]; and a wheeled walker type [3,35]. According to the studies, the walker study by Morone et al. [35] and the whole-body suit study by Monticone et al. [34] were more effective than conventional therapy. However, other studies with exoskeletons did not report more therapeutic effects than conventional therapy for subacute stroke patients [3,16,28,32]. The end-effector type freely guides stepping movements throughout the proximal part of the lower extremity while fixing the distal part of the lower extremity [60]. The RCTs in this review conducted the end effectors to support the above ankle and to facilitate the below ankle movements [31,33]. The end-effector type of a robot-assistive device was a volitional ankle movement control system [31] and a seating-type robot system [33]. The two RCTs reported that robot-assistive training showed more therapeutic effects than conventional therapy. Therefore, regardless of the type of robot-assisted device, qualitative synthesis cannot prove whether it is more effective than conventional therapy for postural stability and gait function in subacute stroke patients.

The review performed quantitative synthesis based on the participant’s information. In this review, four variables were quantitatively synthesized in the RCTs of subacute stroke patients. As a result of the synthesis, the BBS, which evaluates static and dynamic postural balance through 14 predetermined tasks regarding mobility, showed significant therapeutic benefits after robot-assistive training compared with conventional therapy [3,32,33,34]. Postural stability means an even weight bearing on both feet. It decreases due to weight-bearing asymmetry, muscle weakness, bothered perception, devastated ankle proprioception, cognitive impairment, and visual dependency after a stroke, which restricts mobility and functional independence [61]. An improvement in postural stability means that the prerequisites for improving the mobility and functional independence of subacute stroke patients have been fulfilled by robot-assistive training.

The RCTs for chronic stroke patients also used the exoskeleton type [38,39,42,44,48,49,50,51,54], single-segment supporter [46,47], and end-effector type [52] of robot-assistive training. The RCTs reported positive effects compared with conventional therapy; five RCTs used an exoskeleton type [38,39,42,50,51], and two RCTs used a single-segment supporter [46,47]. In quantitative synthesis, two variables, BBS [38,47,48,49,52] and FAC [47,48,49], showed significant positive effects after robot-assistive training for chronic stroke patients. Based on BBS, robot-assistive training was beneficial for postural stability in chronic and subacute patients. Positive efficacy was not proved in other variables; however, for TUG and 6mWT, it is also essential to show an effect in FAC. The result of FAC synthesis will be a signal to prove that robot-assistive training is effective in improving the gait function of chronic stroke patients. Robotics technology developed around exoskeletons and end effectors has been used for treating patients with a stroke for the past 20 years. However, technology development continues to prove its effectiveness and replace human efforts [50,62]. In this review, the response that some research is effective is probably the result of continuing research with robot-assistive technology.

AFO is the most common orthosis used for stroke patients. Notably, few studies have individually investigated the effects of AFO; however, efforts to prove the effects and develop new devices are continuing [41,45]. Two RCTs for chronic stroke patients reported that the assistive device showed benefits compared with conventional therapy [41,45], but the RCTs did not involve a meta-analysis. In the review, two RCTs involved ankle or toe spreaders [40,55], one RCT involved postural insoles [43], and one RCT involved non-elastic taping [53]. An ankle spreader and postural insoles showed positive effects, but a toe spreader and non-elastic taping did not. It is not desirable to conclude the effectiveness of assistive devices from one or two RCTs. The studies suggest that future research about assistive devices should be continued.

Gait function from daily living to community ambulation has major implications for health conditions. For stroke survivors, it is a vital predictor for an individual’s independence, social integration, and quality of life [1]. Assistive technology devices, specifically robotics, have benefits regarding postural stability and gait function to solve a clinical problem that often remains compromised in terms of daily mobility and community ambulation for stroke survivors. In the future, assistive technology, from simple walking aids to a wide variety of other high technologies like robotics, will be investigated for the potential to reduce residual disability, slow functional declines and lower health care costs, and decrease the burden of care regarding postural stability and gait function for stroke survivors. The same variables were used among the RCTs included in the review in this study; however, a meta-analysis could not be performed due to a lack of participant information. Some studies described results only in graphs, not Arabic numerals, and some studies provided outcome measures only as changeable values from pre-training to post-training. In the future, researchers must provide participant information in essential numerals for readers and the citation of future studies when conducting research.

## 5. Conclusions

This review aimed to present clinical evidence regarding the most frequently utilized assistive devices and to provide a comprehensive summary of the therapeutic effects of assistive technology devices on postural stability and gait function in patients with a stroke. The findings of this review indicated that the most frequently conducted assistive technology device for patients with a subacute and chronic stroke was rehabilitative robotics for postural stability and gait. The robot-assistive training showed beneficial therapeutic effects for postural stability in subacute and chronic stroke patients compared with conventional therapy. The robot-assistive training showed the potential to improve the gait function in chronic stroke patients compared with conventional therapy. The robot-assistive technology devices can be attributed to assisting experts, facilitating more movement repetitions and physical capacities, lowering human resources, and encouraging the participants within a given timeframe. This review suggests that robot-assistive technology devices will be used in rehabilitative approaches for postural stability and gait function in subacute and chronic stroke patients.

## Figures and Tables

**Figure 1 healthcare-11-02225-f001:**
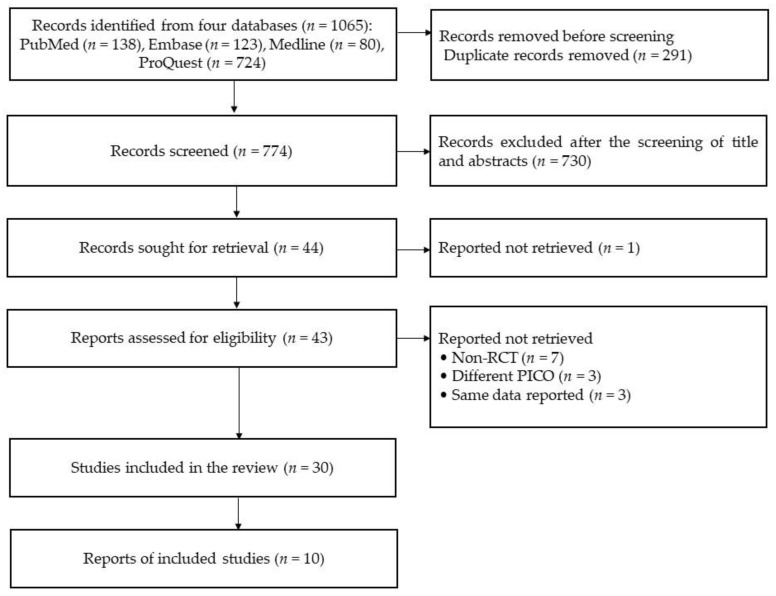
The diagram of this systematic review (source from Page et al. [22]).

**Figure 2 healthcare-11-02225-f002:**
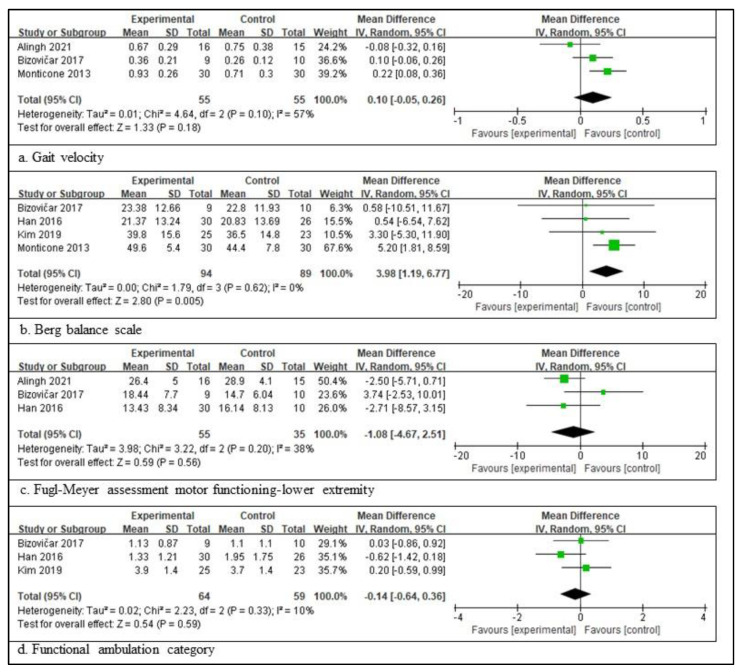
Outcome measures to examine the assistive technology devices for postural stability and gait function for subacute stroke [3,28,31,32,33,34]. The size of the square is proportional to the weight of the study with the pooled estimate, and the line in the middle of the square is the confidence interval for each study. The green color of the block means that the data are continuous. The placement of the center of the diamond on the x-axis represents the point estimate, and the width of the diamond represents the 95% CI around the point estimate of the pooled effect [23].

**Figure 3 healthcare-11-02225-f003:**
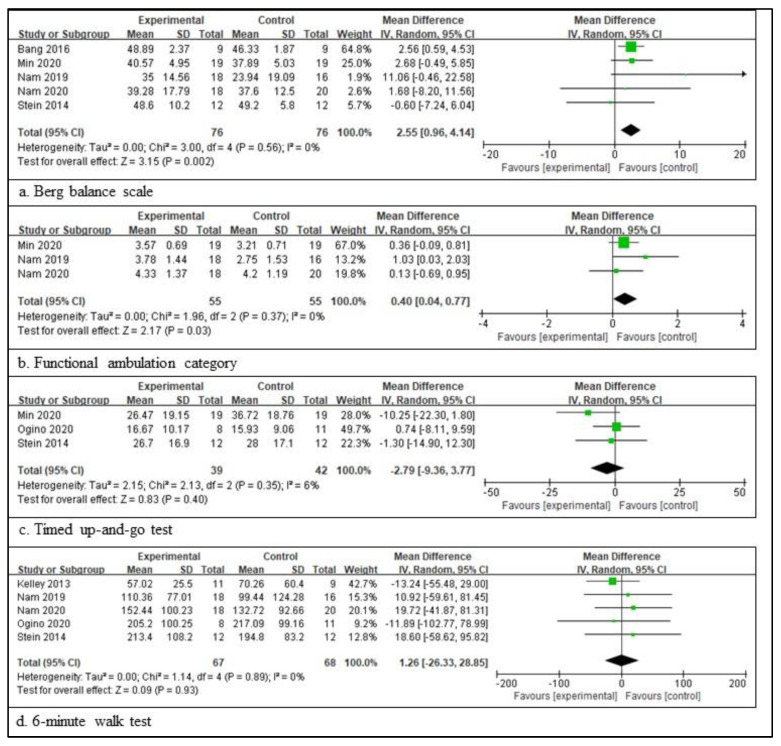
Outcome measures to examine the assistive technology devices for postural stability and gait function for chronic stroke [38,44,47,48,49,50,52]. The meaning of the figure (squares and diamonds) is the same as that of Figure 2.

**Table 1 healthcare-11-02225-t001:** A qualitative synthesis of the selected studies in the review of robotics for subacute patients.

Author; Year; Nation	Participants; Age	Intervention	Device Type	Intensity	Comparison	Outcome	Additional Therapy	Summary of Findings
Alingh; 2021; the Netherlands [28]	32 (female, 12) EG: 60.6 ± 9.3 CG: 56.8 ± 9.8	Robot-assisted gait training with BWSS	MOOG BV	30 min/session 18 sessions	Conventional gait training	Gait speed, step width, step length, single-support time, 10MWT, 6mWT, FGA, TUG, FMA-LE, MI	Conventional gait training 30 min 12 sessions	EG = CG
Bizovičar; 2017; Slovenia [3]	19 (female, 11) EG: 52 ± 8 CG: 60 ± 10	Walking training with an E-go device	E-go device (motorized wheels)	45 min/session 15 sessions	Conventional physiotherapy	Walking speed, walking distance BBS, FMA-LE, FAC	Standard physiotherapy training	EG = CG
Forrester; 2014; USA [31]	34 (female, NR) EG: 63.3 ± 2.3 CG: 60.0 ± 3.1	Robot training	Anklebot	60 min	Manual stretching	FIM, gait velocity, BBS, AROM, MMT, STG parameters, PAV, MAV, normalized jerk, target success	Usual physical therapy	EG > CG
Han; 2016; South Korea [32]	56 (female, 24) EG: 67.89 ± 14.96 CG: 63.20 ± 10.62	Robot-assistive gait training	Lokomat	30 min/session 20 sessions	Conventional therapy	K-MBI, BBS, FAC, FMA-LE, baPWV	Rehabilitation therapy 30 min	EG = CG, except for baPWV (EG > CG)
Kim; 2019; South Korea [33]	48 (female, 15) EG: 57.7 ± 12.9 CG: 60.4 ± 13.2	Robot-assisted gait training	Morning Walk^®^	30 min 15 sessions	Conventional therapy	FAC, MI, 10MWT, MBI, RMI, BBS	Conventional therapy 60 min	EG > CG on affected lower limb’s strength and postural balance
Monticone; 2013; Italy [34]	60 (female, 26) EG: 62.1 ± 9.7 CG: 60.2 ± 6.1	Robot training	Regent Suit	30 min 20 sessions	Usual care	Gait speed, cadence, step length, symmetry index FIM, BI	Neuromotor rehabilitation 60 min/session 20 sessions	EG > CG
Morone; 2016; Italy [35]	44 (female, 14) EG: 61.50 ± 10.97 CG: 64.09 ± 16.27	Conventional walking training using a servo-assistive robotic walker	i-Walker	40 min 20 sessions	Conventional walking-oriented therapy	6mWT, 10MWT, Tinetti’s scale, MAS, BI, CNS	Hand therapy	EG > CG
van Nunen; 2015; the Netherlands [16]	30 (female, 14) EG: 50.0 ± 9.6 CG: 56.0 ± 8.7	Robot-assisted treadmill training	Lokomat	2 h per week/8 weeks	Conventional therapy	Walking speed, FAC, BBS, RMI, FMA-LE, TUG	Conventional therapy 1.5 h per week/8 weeks	EG = CG

baPWV, brachial-ankle pulse wave velocity; BBS, Berg balance scale; BI, Barthel index; CNS, Canadian neurological scale; FAC, functional ambulatory category; FIM, functional independence measure; FMA-LE, Fugl-Meyer assessment of motor functioning—lower extremity; K-MBI, Korean version of modified Barthel index; MAS, modified Ashworth scale; MAV, mean angular velocity; MI, motricity index; PAV, peak angular velocity; RMI, Rivermead mobility index; TUG, timed up-and-go test.

**Table 2 healthcare-11-02225-t002:** A qualitative synthesis of the selected studies in the review of other technology for subacute patients.

Author; Year; Nation	Participants; Age	Intervention	Device Type	Intensity	Comparison	Outcome	Additional Therapy	Summary of Findings
Choi; 2020; South Korea [29]	36 (female, 16) EG1: 64.1 ± 10.5 EG2: 62.4 ± 12.1 CG: 67.4 ± 12.9	Trunk stabilization exercise with pelvic compressor belt	Compressor belt	30 min/session 30 sessions	Comprehensive rehabilitation therapy	PASS, TUG, COL, LOS	Neurodevelopmental therapy 60 min/session 30 sessions	EG1 > EG2 = CG
de Seze; 2011; France [30]	28 (female, 10) EG: 56.4 ± 8 CG: 53 ± 13	Ankle–foot orthosis	Chignon ankle–foot orthosis	30 days	Standard ankle–foot orthosis	10MWT, FAC, PASS, FIM	Not available	EG > CG
Nikamp; 2017; the Netherlands [36]	33 (female, 13) EG: 56.9 ± 9.6 CG: 57.5 ± 9.1	Early provision	Ankle–foot orthosis: polyethylene, semi-rigid, and rigid	26 weeks	Delayed provision	10MWT, BBS, FAC, 6mWT, TUG, RMI, BI, MI, stairs test	Not available	EG = CG
Yamamoto; 2018; Japan [37]	40 (female, 4) EG: 59.2 ± 9.8 CG: 60.2 ± 12.3	Gait training with ankle–foot orthosis	Ankle–foot orthosis with plantar flexion stop	1 h/session 14 sessions	Ankle–foot orthosis with plantar flexion resistance	Gait velocity, loading response times, single-stance times, pre-swing time, swing time, ground reaction forces, COP progression, joint angle, moment, and power at ankle, knee, and hip Pelvic tilt, thoracic tilt	Not available	EG = CG

10MWT, 10 m walking test; BBS, Berg balance scale; BI, Barthel index; COL, center of loading; COP, center of pressure; FAC, functional ambulatory category; FIM, functional independence measure; LOS, limit of stability; PASS, postural assessment scale for stroke; RMI, Rivermead mobility index; TUG, timed up-and-go test.

**Table 3 healthcare-11-02225-t003:** A qualitative synthesis of the selected studies in the review of robotics for chronic patients.

Author; Year; Nation	Participants; Age	Intervention	Device Type	Intensity	Comparison	Outcome	Additional Therapy	Summary of Findings
Bang; 2016; South Korea [38]	18 (female, 9) EG: 53.56 ± 3.94 CG: 53.67 ± 2.83	Robot-assisted gait training	Lokomat	60 min/session 20 sessions	Treadmill training	ABC, BBS, gait velocity, cadence step length, double-support period	Not available	EG > CG
Calabro; 2018; Italy [39]	40 (female, 17) EG: 69 ± 4 CG: 67 ± 6	Robot-assisted gait training	Ekso^TM^	45 min/session 40 sessions	Overground gait training	10MWT, RMI, TUG, GQI, GCD, cadence, stance/swing ratio, sEMG (RF, BF, S, TA), EEG	Conventional physiotherapy training 60min/day	EG > CG
Erbil; 2018; Turkey [42]	43 (female, 27) EG: 50.1 ± 11.8 CG: 48.7 ± 10.4	Robot-assisted gait training	RoboGait^®^	30 min/session	Physical therapy	BBS, TUG, RVGA, MAS, Tardieu scale	Physical therapy 60 min/day	EG > CG
Kelley; 2013; USA [44]	20 (female, 7) EG: 66.91 ± 8.50 CG: 64.33 ± 10.91	Robotic-assisted locomotor training	Locomat^®^	60 min/session 40 sessions	Overground training	10MWT, 6MWT, FIM-LE, Barthel index, SIS	Not available	EG = CG
Lee; 2019; South Korea [46]	26 (female, 12) EG: 61.85 ± 7.87 CG: 62.25 ± 6.36	Gait enhancing and motivating system on five treadmill sessions and five overground sessions	Wearable hip-assistance robot	45 min/session 10 sessions	Non-robot on five treadmill sessions and five overground sessions	MAS, K-MoCA, FAC, gait velocity, cadence, stride length, temporal symmetry ratio, spatial step symmetry ratio, muscle effort symmetry ratio	Not available	EG > CG
Min; 2020; South Korea [47]	38 (female, 14) EG: 61.47 ± 11.15 CG: 56.36 ± 9.16	Robot training	Trunk stability rehabilitation robot trainer (3DBT-33)	30 min/session 20 sessions	Conventional physical therapy	BBS, TUG, FMA-LE, FAC, K-MBI	Conventional physical therapy 30 min/day	EG > CG
Ogino; 2020; Japan [50]	19 (female, 4) EG: 66.1 ± 9.6 CG: 65.0 ± 7.7	Robot-assisted gait training	GEAR system (gait exercise assistive robot)	40 min/session 20 sessions	Treadmill training	10MWT, TUG, SF-8, 6mWT, GRC	Conventional therapy 20 min/day	EG > CG
Rodrigues; 2017; USA [51]	18 (female, 8) EG: 50.6 ± 14.4 CG: 59.3 ± 13.8	Robot-assisted BWSTT	Robot-assisted (slow)	30 min/session	Robot-assisted (fast)	FAC, TUG, 6mWT, 10MWT, BBS, FMA-LE	Not available	EG > CG
Stein; 2014; USA [52]	24 (female) EG: 57.6 ± 10.7 CG: 56.6 ± 15.1	Robot training	Bionic leg device	60 min/session 18 sessions	Group exercise	10MWT, 6mWT, 5XSST, TUG, BBS, CAFÉ 40, EFAP, Romberg	Not available	EG = CG
Wu; 2014; USA [54]	28 (female, 10) EG: 53.6 ± 8.9 CG: 57.4 ± 9.8	Robotic gait training applied to the paretic leg to assist with leg swing	Custom-designed, cable-driven robotic gait training system	45 min/session 18 sessions	Robotic gait training applied to the paretic leg to assist with leg swing	SLS, step length, step asymmetry, cadence ABC, SF-36, MAS	Not available	EG = CG

5XSST, 5-time sit-to-stand test; 6mWT, 6 min walking test; 10MWT, 10 m walking test; ABC, activities-specific balance confidence scale; BBS, Berg balance scale; BF, biceps femoris; CAFÉ 40, California functional evaluation 40; EFAP, Emory functional ambulation profile; FAC, functional ambulation category; GCD, gait cycle duration; GRC, global rating of change; GQI, gait quality index; MAS, modified Ashworth scale; K-MBI, Korean version of modified Barthel index; K-MoCA, Korean version of Montreal cognitive assessment; MI, motricity index; RF, rectus femoris; RVGA; S, soleus; SF-36, 36-item short-form survey; SF-8, medical outcome study 8-item short-form health survey; SIS, stroke impact scale; TA, tibialis anterior; TUG, timed up-and-go test.

**Table 4 healthcare-11-02225-t004:** A qualitative synthesis of the selected studies in the review of other technology for chronic patients.

Author; Year; Nation	Participants; Age	Intervention	Device Type	Intensity	Comparison	Outcome	Additional Therapy	Summary of Findings
Chiong; 2013; Singapore [40]	9 (female, 4) 48 (34–61)	Toe flexor stretches with toe spreader	Rolyan Ezemix elastomer putty with sports sandals	Ambulated during the 6-month study period	Toe flexor stretches	Gait velocity, stride length, step length, plantar surface contact area, MAS, BBS, VAS	Not available	EG = CG
Cho; 2022; South Korea [41]	30 (female, 7) EG: 51.8 ± 12.0 CG: 55.0 ± 10.9	Passive biaxial ankle movement training with electrical stimulation therapy	Ankle movement system	40 min/session 20 sessions	Electrical stimulation therapy	Proprioception, pROM, strength, FMA-LE, BBS, TUG, FES, walking speed, step length, step time, step width, ROM	Inpatient rehabilitation program	EG > CG on pROM and strength
Ferreira; 2017; Brazil [43]	20 (female, 4) EG: 59.2 ± 10.4 CG: 60.3 ± 13.3	Postural insoles for equinovarus foot	Postural insoles	3 months	Placebo insoles without corrective elements	Stance phase, swing phase, double support, step length, stride length, mean velocity, cadence, sagittal kinematic plots	Conventional physical therapy	EG > CG on dorsiflexion and knee flexion at post-training
Kluding; 2012; USA [45]	197 (female, 79) EG: 60.71 ± 12.24 CG: 61.58 ± 10.98	Gait training with foot drop stimulator	NESS L300 Foot Drop System	30 weeks	Gait training with ankle–foot orthosis	Walking speed, FMA, 6mWT, FRT, SIS, BBS, FAC	Gait training and home exercise program Eight sessions	EG > CG on user satisfaction
Nam; 2019; South Korea [48]	34 (female, 17) EG: 48.33 ± 15.56 CG: 68.56 ± 17.35	Electromechanical-assisted gait training	Exowalk, an electromechanical exoskeleton	30 min/session 20 sessions	Physical therapy-assisted gait training	FAC, RMI, 10MWT, 6mWT, MI, BBS, MBI	Physical therapy	EG = CG
Nam; 2020; South Korea [49]	38 (female, 16) EG: 60.00 ± 11.48 CG: 57.30 ± 8.71	Electromechanical-assisted gait training	Exowalk, an electromechanical exoskeleton	60 min/session 20 sessions	Physical therapy-assisted gait training	FAC, RMI, 10MWT, 6mWT, MI, BBS, MBI	Physical therapy	EG = CG
Wang; 2022; Taiwan [53]	21 (female, 6) EG: 62.27 ± 10.10 CG: 63.30 ± 7.05	Non-elastic hip taping combined with gait training	Non-elastic taping	50 min/session 12 sessions	Sham	Gait velocity, double-support time, spatial symmetry index, temporal symmetry index, BBS, 6mWT, FES	Not available	EG = CG
Yoo; 2018; South Korea [55]	16 (female, 5) EG: 58.5 ± 9.4 CG: 53.9 ± 6.0	Ankle stretching exercise	Motorized ankle stretcher	Seven sessions	Stretching board exercise	Ankle ROM, SOT, walking speed, cadence, step length	Not available	EG > CG

6mWT, 6 min walking test; 10MWT, 10 m walking test; ABC, activities-specific balance confidence scale; BBS, Berg balance scale; FAC, functional ambulation category; FES, fall efficacy scale; MBI, modified Barthel index; MI, motricity index; pROM, passive range of motion; SIS, stroke impact scale; SOT, sensory organization test; TUG, timed up-and-go test; VAS, visual analogue scale.

## Data Availability

The data presented in this study are available on request from the corresponding author.

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
