# Peer review of "Assistive Technology Involving Postural Control and Gait Performance for Adults with Stroke: A Systematic Review and Meta-Analysis"

_healthcare, 2023, doi:10.3390/healthcare11152225_

Round 1

Reviewer 1 Report

Congratulations for your article, it is a particularly novel review. However, it is necessary to make some clarifications:

- At the introductory level and in the discussion section, it would be possible to go more deeply into the fundamental aspects of physiotherapy treatment. 

- The design of the review (section 2.1) indicates that PRISMA guidelines are followed, but the diagram shown in Figure 1 does not fully represent the PRISMA model. 

- The tables need to be revised in terms of distribution and content. The tables are long and there are words that are cut off in the line.

- It might be interesting to include an image of the devices being evaluated.

- The manuscript is complete and includes the analysis of many variables; it is necessary to condense the information and highlight the most important points for the reader.

Author Response

RESPONSES TO THE REVEIWER #1

Congratulations for your article, it is a particularly novel review. However, it is necessary to make some clarifications:

Thank you for your generous suggestion. You have carefully pointed out the errors we have made in writing the study. Your suggestion makes us more scientific view. We have revised the study according to your suggestions. Below is a response to your suggestion.

- At the introductory level and in the discussion section, it would be possible to go more deeply into the fundamental aspects of physiotherapy treatment. 

Response: We corrected the introduction and discussion sections. Please see the revised manuscript.

- The design of the review (section 2.1) indicates that PRISMA guidelines are followed, but the diagram shown in Figure 1 does not fully represent the PRISMA model. 

Response: We corrected the diagram, according to your suggestion.

- The tables need to be revised in terms of distribution and content. The tables are long and there are words that are cut off in the line.

Response: According to your suggestion, we separated the tables.

- It might be interesting to include an image of the devices being evaluated.

Response: That’s right. This study was a systematic review and we did not use a specific assistive device to correct the data, so it is difficult to use an image.

- The manuscript is complete and includes the analysis of many variables; it is necessary to condense the information and highlight the most important points for the reader.

Response: We known that. This review aimed all the assistive devices used to improve postural control and gait performance, not one assistive device, so the contents are vast. However, according to your suggestion, we narrowed down the discussion section.

Reviewer 2 Report

Dear Authors,

thank you for giving me the opportunity to review this study. The work deals with the efficiency of gait and balance rehabilitation supported by robotic devices compared to the classical methodology in the stroke patient. The article is interesting and well structured. The only indication I would give is the following: it would be interesting to see if the positive effects of robot-assisted rehabilitation are confirmed by a computerized evaluation methodology (Gaitrite, Stereophotogrammetry, Balance Platforms ecc.). This would give further confirmation of what has been reported; BBS and FAC, although they are validated scales and used in clinical practice, can potentially be influenced by the human error component; in the field of research it would be advisable to seek an instrumental confirmation of the results.

Author Response

RESPONSES TO THE REVEIWER #2

Thank you for your generous suggestion. You have carefully pointed out the errors we have made in writing the study. Your suggestion makes us more scientific view. We have revised the study according to your suggestions. Below is a response to your suggestion.

Dear Authors,

thank you for giving me the opportunity to review this study. The work deals with the efficiency of gait and balance rehabilitation supported by robotic devices compared to the classical methodology in the stroke patient. The article is interesting and well structured. The only indication I would give is the following: it would be interesting to see if the positive effects of robot-assisted rehabilitation are confirmed by a computerized evaluation methodology (Gaitrite, Stereophotogrammetry, Balance Platforms ecc.).This would give further confirmation of what has been reported; BBS and FAC, although they are validated scales and used in clinical practice, can potentially be influenced by the human error component; in the field of research it would be advisable to seek an instrumental confirmation of the results.

Response: Thank you for your generous suggestion. Your suggestion make us more scientific view. This study performed quantitative synthesis when at least three studies had relevant data and sufficient homogeneity in population, therapeutic interventions and outcome measures. Unfortunately, fewer than three RCTs had been reported the relevant data of the spatiotemporal parameters.

Reviewer 3 Report

Data analysis: Was the statistical heterogeneity across the studies determined only by I2? When the p-value and I2 statistics are examined from the report (Figures 2 and 3); the p-values from the Chi-Square test are higher than 0.05/0.10 for most of the variables. Similarly, the I2 statistics are lower than the percentage at line 145, except 57% in Figure 2a. As a result of which statistical finding did you make the inference in line 246 and line 272?

Line 145: “An I2 >40%...” will be “An I2 >40%...”. It would be good to add a reference to the given limit value (40%).

Line 154: According to the information given in the method, the reference lists of all publications determined in the four search engines were also examined. However, I could not see information about whether any studies were added or not from these lists.

Line 157: 82+194+448= 724? Please check.

Line 160: 7+3+3+1=14 but 17 according to Figure 1. Please check.

Line 161 and Figure 1: 43– 14 (according to text and 17 according to Figure 1?) = 29 studies. But 30 in Figure and text. Please check.

Table 1-2-Figure 2: Adding the reference numbers of the articles in Tables 1-2 and Figure 2 will facilitate the readers to match the reports given in the text with the tables.

Moderate editing of English language required

Author Response

RESPONSES TO THE REVEIWER #3

Thank you for your generous suggestion. You have carefully pointed out the errors we have made in writing the study. Your suggestion makes us more scientific view. We have revised the study according to your suggestions. Below is a response to your suggestion.

Data analysis: Was the statistical heterogeneity across the studies determined only by I2? When the p-value and I2 statistics are examined from the report (Figures 2 and 3); the p-values from the Chi-Square test are higher than 0.05/0.10 for most of the variables. Similarly, the I2 statistics are lower than the percentage at line 145, except 57% in Figure 2a. As a result of which statistical finding did you make the inference in line 246 and line 272?

Response:

Line 145: “An I2 >40%...” will be “An I2 >40%...”. It would be good to add a reference to the given limit value (40%).

Response: We corrected that from I2 to I2 and added that reference.

Line 154: According to the information given in the method, the reference lists of all publications determined in the four search engines were also examined. However, I could not see information about whether any studies were added or not from these lists.

Response: We added that.

Line 157: 82+194+448= 724? Please check.

Response: We checked that according to your comment.

Line 160: 7+3+3+1=14 but 17 according to Figure 1. Please check.

Response: We checked that according to your comment.

Line 161 and Figure 1: 43– 14 (according to text and 17 according to Figure 1?) = 29 studies. But 30 in Figure and text. Please check.

Response: We checked that according to your comment.

Table 1-2-Figure 2: Adding the reference numbers of the articles in Tables 1-2 and Figure 2 will facilitate the readers to match the reports given in the text with the tables.

Response: According to your suggestion, we added that.